# Sports Games and Motor Skills in Children, Adolescents and Youth with Intellectual Disabilities

**DOI:** 10.3390/children10060912

**Published:** 2023-05-23

**Authors:** Nikola Aksović, Tatiana Dobrescu, Saša Bubanj, Bojan Bjelica, Filip Milanović, Miodrag Kocić, Milan Zelenović, Marko Radenković, Filip Nurkić, Dejan Nikolić, Jovan Marković, Milena Tomović, Ana-Maria Vulpe

**Affiliations:** 1Faculty of Sport and Physical Education, University of Priština, 38218 Priština, Serbia; kokir87np@gmail.com; 2Department of Physical Education and Sport Performance, Vasile Alecsandri University, 600115 Bacau, Romania; zaharia.ana@ub.ro; 3Faculty of Sport and Physical Education, University of Niš, 18000 Niš, Serbia; bubanjsale@gmail.com (S.B.); miodrag.kocic73@gmail.com (M.K.); radenkom9@gmail.com (M.R.); filipnurkic@gmail.com (F.N.); 4Faculty of Medicine, University of Belgrade, 11000 Belgrade, Serbia; denikol27@gmail.com; 5Faculty of Physical Education and Sports, University of East Sarajevo, 71126 Lukavica, Bosnia and Herzegovina; vipbjelica@gmail.com (B.B.); milanzeleni13@gmail.com (M.Z.); 6University Children’s Hospital, 11000 Belgrade, Serbia; 7Faculty of Pedagogy, University of Kragujevac, 31000 Užice, Serbia; jovo.markovic.72@gmail.com; 8Sports Medicine Laboratory, Department of Physical Education and Sports Science, Aristotle University of Thessaloniki, 54124 Thessaloniki, Greece; milenatomovic83@gmail.com

**Keywords:** basketball, football, intellectual disability, impact, down syndrome

## Abstract

(1) Background: Sports games are one of the best ways of engaging in physical activity for individuals with intellectual disabilities (ID) and Down syndrome (DS). This systematic review of the current literature aims to identify and sum relevant data on motor skills and clarify whether there are positive effects of sports programs in motor skills games in children, adolescents, and youth with ID and DS. (2) Methods: The systematic review of the papers was carried out following the methodological guidelines and by the Preferred Reporting Items for Systematic Reviews and Meta-Analyses (PRISMA) consensus. The following electronic databases were researched: PubMed, MEDLINE, Google Scholar, ScienceDirect, and ERIC from 2001 to 2023. (3) Results: The basketball test battery can be used to improve and monitor basketball training. Basketball players with lower levels of ID achieved better results, especially those with disabilities of the II and III degrees. Futsal has a positive impact on the coordination, and the differences between the test results of the vertical jump with and without an arm swing, that can be seen indirectly as a coordination factor, were statistically significant. (4) Conclusions: Basketball is recommended as an effective and practical rehabilitation program for children, adolescents, and youth with ID and DS. Futsal is an interesting and helpful activity for individuals with ID as well.

## 1. Introduction

An intellectual disability (ID) is defined as a condition where the individual is physically impaired or physiologically underdeveloped, which is especially characterized by the disruption of those abilities during the developmental period and contributes to a general level of intelligence and speech and cognitive, motor, and social abilities [1]. ID is generally defined as having an IQ score below 70, along with limitations in adaptive functioning (i.e., the ability to learn and apply skills, solve problems, and adapt to new situations), which can significantly impact an individual’s daily living skills, communication abilities, and social interactions. There are different levels of intellectual disability, ranging from mild, moderate, severe to profound, depending on the degree of impairment in cognitive and adaptive functioning [2]. Numerous diseases and conditions are associated with various degrees of intellectual disabilities. Within the term ID, there are several conditions, such as Down syndrome (DS), Turner syndrome, Klinefelter syndrome, phenylketonuria, fragile X-chromosome syndrome, Rett syndrome, autism, childhood cerebral palsy, and many others [3,4,5,6,7,8,9,10]. Approximately 2% of children in the general population have some degree of ID, while this percentage is reported to be up to 25% among children with cerebral palsy [11]. In children with epilepsy, ID occurs in around 9% of cases [12].

ID is a condition of delayed or incompletely developed mental abilities until the age of 18, whether caused by hereditary factors or due to illness [13]. The developmental characteristics of this population are common in all spheres and primarily expressed in cognition deficits, reduced levels of adaptive behavior, and slow motor development [14,15]. As a result, motor test results of adolescents with an ID attending a special high school lag three to four years behind in comparison to adolescents attending a regular high school [16]. Children with ID have low levels of physical fitness relative to their peers without ID. Coordination, balance, endurance, muscle power, explosive power, and sprinting are particularly important for the general motor behavior in individuals with ID [17,18,19]. Numerous studies indicate that children with ID have low levels of cardiovascular endurance, muscle strength, muscular endurance, speed, balance, and agility [20,21,22,23,24]. Rimmer [25] indicated the problem of an increase in physical inactivity and obesity in individuals with ID, recommending professionals begin with various health promotions, including greater involvement in physical activity. Additionally, it should be said that children with ID are socially challenged and want to isolate themselves, which may be a contributor to their physical inactivity [21]. Thus, physical activity in children and adolescents with ID leads to motor skills development, and one of the best-organized activities is sports games.

Sports games are one of the best ways of engaging children and adolescents with disabilities in physical activity, because sports are considered a product of physical and cognitive potential [26,27]. Physical potential includes physical fitness, while cognitive potential includes intelligence, as a multidimensional area, including reasoning, planning, decision-making, rapid learning, and learning from experience [28]. Sport helps athletes with ID to increase their self-confidence and play a key role in socializing and collaborating with people with some degree of illness [29]. As far as team sports are concerned, basketball is very popular in sports and physical education programs because it implies different physical capacities and significantly contributes to the development of these individuals [30], especially for children and individuals with ID [31,32]. Basketball is a very popular activity and sports education program for people with ID because it implies motor skills [32] such as running, jumping, kicking, passing, and running the ball [29,31].

DS is a congenital autosomal anomaly that occurs due to a trisomy of chromosome 21 [33,34]. Numerous studies have indicated that muscle strength is considered a significant fitness parameter in individuals with DS [6,35,36,37]. Carmeli and associates [38] compared the muscle strength of individuals with ID and those with DS and indicated that people with DS and some other IDs have a lower level of muscle strength than individuals with only an ID. The muscular strength of the upper and lower extremities of individuals with DS is 50% less than in individuals with ID but without DS [36,39,40]. Muscle strength, endurance, and dynamic stability in individuals with DS are considered to be significant factors in a good quality of life and functional independence [41]. Some studies suggested that progressive resistance training can be used to develop muscle strength in children, adolescents, and individuals with DS [40,42,43], while other studies [44,45] indicated that this training method increases the risk of disabilities.

This review aims to present the effects of sports games on motor skills in children, adolescents, and youth with ID and DS. Thus, this systematic review of the current literature aims to identify and sum the relevant data on motor skills and clarify whether there are positive effects of sports programs in motor skills games in children, adolescents, and youth with ID and DS.

## 2. Materials and Methods

The systematic review of the papers was carried out following the methodological guidelines and by the Preferred Reporting Items for Systematic Reviews and Meta-Analyses (PRISMA) consensus [46].

### 2.1. Search Strategy 

The following electronic databases were searched: PubMed, MEDLINE, Google Scholar, ScienceDirect, and ERIC from 2001 to 2023. The search was carried out based on a combination of keywords: intellectual disabilities, intellectual disability, Down syndrome, sports games, effect, basketball, football, soccer, futsal, and handball. 

All headlines and abstracts were reviewed for the potential papers, which were included in the systematic review. The lists of previous and original research were also reviewed. The relevant studies were obtained when the studies completed the criteria for inclusion after detailed research. Wherever it was possible, the research strategy was modified and adapted to each database research to increase the sensitivity of this review paper. 

### 2.2. Criteria for Inclusion

Type of study:Randomized controlled and non-randomized studies were examined and included in further analysis, while uncontrolled studies were excluded. This review included studies written in Serbian and English.

The sample of participants:Study participants included athletes (amateurs/professionals), children, adolescents, and youth, with and without ID, of both genders and any age (no restriction), regardless of the degree of disability.

Type of intervention:Studies that determine the effects of sports game programs on individuals with ID were examined regardless of the length and type of study.

The type of obtained results:The preliminary results obtained for our systematic review were motor skills after the training program. Studies were included if the impact of sports game programs on the physical fitness abilities of athletes was demonstrated. Secondary results, which are primarily related to the systematic review of papers, consisted of the following variables: psychosocial characteristics, anthropometric characteristics, heart rate, blood pressure, obesity, and cholesterol.

### 2.3. Criteria for Exclusion

Type of study:Studies written in a language which was not Serbian or English;Duplicates;Conference abstracts.

### 2.4. Data Extraction

The collected research projects used for this review are shown in Table. For each study, the following parameters were shown: (1) study characteristics including author (s) and year of publication; (2) information about participants such as sample size, gender, age, and groups; (3) aim of the study; (4) duration of the training program; (5) key findings of the studies obtained by the authors.

## 3. Results

One hundred and two articles were identified from the database search, with an additional six articles identified through reference lists. After removing duplicates and eliminating articles based on title and abstract screening, forty-six studies remained. An evaluation of the remaining forty-six studies was conducted independently by two researchers. Following the final screening process, seventeen studies were included in the systematic review (Table 1). Details of the study selection process are presented in Figure 1.

### 3.1. Characteristics of Participants with Intellectual Disabilities

Ten studies included both male and female participants [47,48,49,50,51,52,53,54,55,56]. 

Five studies included male participants [57,58,59,60,61]. The gender of the participants was not specified in two studies [32,62].

Most studies had adolescents [49,50,51,52,53,56,57,58,59,60,61], youth [47,54], children [32], children and adolescents [55], adolescents and adults [48] or children, and adolescents and adults [62].

### 3.2. Characteristics of the Studies

The duration of the training program ranged from four weeks to four years. The most common duration of the training program was eight weeks (seven studies) [48,49,50,52,53,58,60], six months (three studies) [47,54,62], eight months (two studies) [59,63], thirty-three (one study) [56], seven months (one study) [57], twelve weeks (one study) [55], five weeks (one study) [32] and four weeks (one study) [51].

The most frequent duration of the training sessions was two hours per week (4 × 30 min) [49,50,53], (2 × 60 min) [52,57], and four and a half hours per week (3 × 90 min) [55,56,58,60], then three hours per week (3 × 60 min) [54,63], four hours per week (4 × 60 min) [59], one hour and a half per week (1 × 90 min) [48], and one hour per week (2 × 30 min) [32].

The duration or length of the experimental program and training sessions were not stated in one study [61].

Regarding sports game programs, basketball was the most commonly represented program [32,47,49,50,51,53,57,59,61]. Some studies had basketball players with ID as their sample [47,59]. Studies Özer and associates [63] and Baran and associates [60] identified the effects of Special Olympics (SO) Unified Sports Soccer (UNS) programs on physical fitness, football skills, and psychosocial characteristics of anthropometry in children athletes with and without ID, while the study of Niemeier and associates [48] evaluated the effectiveness of SO Fit 5 health program in improving health measures for individuals with ID. Maano and associates [57] aimed to analyze the effects of alternative sports competitions. Ilkim and Akyol [55] assessed the effects of a table tennis exercise program on reaction times in children with DS. Naczk and associates [56] estimated the influence of a thirty-three-week swimming program on aerobic capacity, physical fitness, level of adjustment and function in water, and body composition of adolescents with DS. Further details of the included studies are shown in Table 1.

**Table 1 children-10-00912-t001:** The systematic review and characteristics of the included studies.

Study/ Country	Participants	Type of Intervention	The Aim of the Study	Training Program	Assessment Tools	Key Findings
Franciosi et al. (2012) [59] / Italy	Male adolescent and adult basketball players with different levels of ID: mild (15%), moderate (54%), severe (29%), and profound (2%) (n = 41) EXP1, competitive category (n = 17) and EXP2, pro-category (n = 24) aged (18–45)	EXP1 and EXP2 implemented training program with PRE and POST assessment of four ability levels of increasing difficulty (from I to IV).	To determine whether a basketball battery of tests can assess basketball skills before and after eight months of training in persons with ID, in relation to competitive and pro-categories, and to analyze variations in specific abilities of basketball players with ID.	8 months	Modified basketball tests for players with ID	Competitive basketball players (ID III) ↑ BH, PS, R, SS. Pro-categories basketball (ID II) ↑ BH, R, PS. Level ID (I, II, III) have a statistically significant negative correlation with ID diagnosis, indicating that basketball players with a lower ID level achieve better results. In both groups ↑. Significant differences between categories ID (I, II, III) in all fundamental areas. Basketball test battery (8 months, 4 h per week) could be useful for improving and monitoring basketball training in both categories.
Franciosi (2007) [47] / Italy	Trained youth basketball players with ID (n = 15) EXP1 (n = 3, mild ID, Level I) EXP2 (n = 8, moderate ID, Level II) EXP3 (n = 3, severe ID, Level III) EXP4 (n = 1, profound ID, Level IV) aged (23.5 ± 4.2) male (n = 11) female (n = 4)	EXP1, EXP2, EXP3, and EXP4 implemented basketball training program.	To determine the effects of the training program on basketball skills and psychological status in basketball players with ID during two sports seasons.	6 months	Modified basketball tests for players with ID. Two psychological questionnaires: (a) the perceived physical ability scale; and (b) the task and ego orientation in sport questionnaire.	Basketball training produced a general improvement after 6 months in both sport seasons. Athletes with lower ID obtained higher ability scores. NC difference PQ. This result could be justified by the positive influence of physical activity in persons with ID, who could know better their physical ability through sport experience.
Kocić et al. (2017) [53] / Serbia	Adolescents with mild ID (n = 50) male (n = 27) female (n = 23) EXP (n = 25) aged (15.7 ± 0.9) CON (n = 25) aged (15.9 ± 0.8)	EXP performed a specially adapted basketball training program in addition to regular PE classes that were not held on the same day. CON performed regular PE classes only.	To examine the effects of an adapted basketball training program on the cardiorespiratory fitness and sport skill performance of adolescents with ID.	8 weeks	Six-minute walk test; modified basketball tests for players with ID	NC EXP, CON H, W, BF, EXP ↑ 6MWT NC differences between groups. Adapted basketball training (4 times per week, 25–35 min) is an adequate stimulus for improvement in cardiorespiratory fitness and sport skill performance of adolescents with mild ID.
Stanisić et al. (2012) [49] / Serbia	Adolescents, elementary school students for children with mild ID (n = 12) male (n = 6) female (n = 6) EXP (n = 12) aged (15.5 ± 1.5)	EXP implemented specially adapted training program.	To identify differences specifically in motor skills after a specially designed eight-week basketball training program.	8 weeks	Modified basketball tests for players with ID	↑ BH, RB, PS, SS. NC 20 m sprint, SBJ, FAH, MSIT, MSAR. A specially designed basketball program (8 weeks, 4 times per week, 30 min) contributes to an increase in specific motor skills.
Ince (2017) [52] / Türkiye	Male and female adolescents and adults (n = 23) EXP (n = 12) aged (22.5 ± 5.2) CON (n = 11) aged (19.3 ± 6.2)	EXP implemented training program CON did not attend any activities regularly throughout eight weeks	To determine the effect of 8-week ball handling training program on upper-lower extremity muscular strength of individuals with Down syndrome	8 weeks	Ball handling test; Hand and leg dynamometry; VJ; SBJ; WST	EXP ↑ LST (22.47-30.90), EXP ↑ WST (23.55–30.92), EXP ↑ DHGS (11.35–14.55), NC EXP, CON VJ, SBJ, NDHGS. NC differences between group. The findings of the study have revealed that 8-week ball handling training program (4 times per week, 30 min) made a positive effect on upper-lower extremity strength of individuals with DS
Radenković et al. (2014) [51] / Serbia	Adolescents of both gender with ID (n = 27) EXP (n = 13) CON (n = 14) aged (16–19)	EXP followed a modified program of basketball elements adjusted to the intellectual abilities of the children. CON followed the program designed for special schools that was written by the Ministry of Education of the Republic of Serbia.	To establish the level of influence of the modified program of basketball techniques on motor development (speed, coordination, and accuracy) of adolescents with ID, and to adapt particular segments of basketball and adjust them to suit the needs of these adolescents.	4 weeks	Battery of seven tests of basic motor skills	On IM, NC in the difference between EXP and CON. On FM differences MPUS (*p* < 0.002), MHTG (*p* < 0.047), MTDA (*p* < 0.008) in favor of EXP group. The conclusions have partly confirmed the hypothesis that the elements of basketball technique have a statistically significant impact on the development of certain motor skills.
Aydogan and Demirok (2023) [61] / Cyprus	Male adolescents with DS (n = 4) EXP (n = 4) aged (13–16)	EG followed video modeling, teaching basic basketball skills and performing skill he/she saw and experienced in the video.	To determine the effectiveness of modeling with video when teaching basketball basic movement skills to individuals with DS.	No available data	Video teaching model; basic basketball skills tests	Teaching by video model (10–20 days) is effective in providing individuals with DS basic basketball skills, persistence, and generalizing alongside different environments and different people. Self-confidence and peer relations were strengthened thanks to their active participation in the lessons.
Mohammadi et al. (2022) [32] / Iran	Children with mild or moderate ID EXP1 (n = 11) aged (11.7 ± 1.85) EXP2 (n = 10) aged (11.1 ± 1.96)	EXP1 trained their basketball free throws according to easy-set goals. EXP2 trained their basketball free throws according to difficult-set goals.	To examine the effects of easy goals versus difficult goals on acquisition and retention of basketball free throws in children with ID.	5 weeks	Sport skill learning assessment for individuals with ID.	EXP1 ↑↑ performance, EXP2 failed to improve their performance. Results indicate that EXP1 children with easy goals facilitate the process of sport skill learning.
Cai and Baek (2022) [54] / South Korea	Youth with DS (n = 22) aged (24 ± 6) EXP (n = 11) CON (n = 11) male (n = 18) female (n = 4)	EXP performed basketball training program. CON performed no structured exercises.	To evaluate the effects of the basketball training program on the body composition and functional fitness of youth with DS.	24 weeks	Twelve-body composition and functional fitness tests; basketball technique tests.	EXP > CON ↑, ↑↑ BM, BMI, WC, HC, WH. EXP > CON ↑, ↑↑ MSAR, MCUP, SLJ, SONL, WOBB, PCR, SHD, OMS. EXP1↑ Body composition, flexibility, muscular strength, balance, aerobic capacity, and basketball functional ability.
Stanisić et al. (2012) [50] / Serbia	Adolescents, elementary school students for adolescents with mild ID (n = 12) male (n = 6) female (n = 6) EXP (n = 12) aged (15.5 ± 1.5)	EXP implemented specially adapted training program.	To evaluate the effects of the eight-week specially adapted basketball training program on the physical fitness of adolescents with ID.	8 weeks	Six-minute walk test	↑ 6 MWT (473.7–672.6 m), ↑ HR, 6 MWT (122.1-116.5), NC W, BF. The eight-week specially adapted basketball training program (4 times per week, 30 min) provides an increase in physical fitness and allows for the planning of a comprehensive experimental procedure on the subject.
Maano et al. (2001) [57] / Canada	Male adolescents with ID (n = 32) EXP1 ( n = 8) aged (14.4 ± 0.9) EXP2 ( n = 8) aged (14.2 ± 1.0) CON1 ( n = 8) aged (13.7 ± 0.7) CON2 ( n = 8) aged (13.5 ± 0.5)	EXP1 implemented alternated SO basketball. EXP2 implemented alternated SO running. CON1 implemented adapted physical activity. CON2 were sedentary.	To examine the effects of a competitive alternated sport program and type of sport (basketball versus running) on the domains of perceived competence and general self-worth.	7 months	Harter’s self-perception profile	NC difference between groups in AC and time, and interaction, SA, PA after seven months of training (2 h per week). NC differences in time behavior and interaction between groups. Differences (*p* < 0.005) in GSW time. Differences (*p* < 0.005) between EXP1 having basketball SO training and school competitions, and EXP2 having SO running training and school competitions.
Kasum et al. (2012) [62] **/** Serbia	Children, adolescents, and adults with ID (n = 26) aged (16.7 ± 9.6) First sub-sample EXP1 (n = 14) aged (9.5 ± 1.2) second sub-sample EXP2 (n = 12) aged (26.6 ± 6.8)	EXP1 and EXP2 implemented school futsal program.	To determine whether and how the implementation of a six-month school futsal program effects the height of the jump of the children, adolescents, and adults with ID.	6 months	Vertical jump with and without arm swing	Differences on FM VJAS, VJ in favor of VJAS in younger (*p* = 0.018) and older (*p* = 0.007). Differences on FM VJ ↑ by 2.0 cm, VJAS ↑↑ by 3.9 cm between VJAS and VJ on FM (*p* = 0.002). In younger subjects, VJ on FM had better results by 2.6 cm (*p* = 0.286), VJAS by 5.2 cm (*p* = 0.033); in older subjects, VJ on FM had better results by 1.0 cm (*p* = 0.084), VJAS by 2.3 cm (*p* = 0.561).
Baran et al. (2013) [60] / Türkiye	Male adolescents with ID (n = 38) TRG EXP (n = 23) aged (14.1 ± 1.1) CON (n = 15) aged (14.51 ± 0.8) and without MR (n = 38) TRG EXP (n = 23) aged (13.2 ± 0.7) CON (n = 15) aged (13.7 ± 0.4)	TRG EXP (WID and WOID) implemented training program in addition to regular PE classes. CON regular PE classes only.	To investigate the effects of a SO UNS soccer program on anthropometry, physical fitness and soccer skills of male adolescent athletes with and without ID who participated in a training program.	8 weeks	Brockport physical fitness test; SO soccer skill test	UNS program of the EXP group (8 weeks, 3 times per week, 1.5 h) after regular physical education classes. Men with and without ID showed significantly higher results of physical fitness and soccer skills than the CON group, which did not participate in any sport after regular physical education classes. UNS program was successful in ↑ fitness and soccer skill performance of youth with and without ID.
Ozer et al. (2012) [58] / Türkiye	Male adolescents with ID (n = 38) SO EXP (n = 23) aged (14.5 ± 1.2) CON (n = 15) aged (14.5 ± 0.8) and without ID (n = 38) PARTNER EXP (n = 23) aged (14.1 ± 0.9) CON (n = 15) aged (13.8 ± 0.5)	SO EXP, PARTNER EXP implemented training program in addition to regular PE classes. CON regular PE classes only.	To investigate the effects of a SO UNS soccer program on psycho-social attributes of adolescents with and without ID.	8 weeks	Friendship Activity Scale; adjective checklist; children’s behavior checklist	UNS program effective in ↓ the problem behaviors of youth with ID and ↑ their SC and FAS scores. UNS program is effective in improving the attitude of youth without ID toward participants with ID. The soccer training program (8 weeks, 3 times per week, 1.5 h per session) is an effective tool in addition to school PE.
Niemeier et al. (2021) [48] / United States	Male and female adolescents and adults with ID EXP (n = 32) aged (20–59) CON (n = 34) aged (18–61)	EXP implemented Fit 5 SO health program. CON attended regular activities.	To evaluate the effectiveness of Fit 5 SO health program in improving health measures for individuals with ID.	8 weeks	Cardiovascular health tests	EXP ↑↑ RHR, DP, BMI CON ↑↑ BMI. The SO Fit 5 program positively impacts RHR and DP, and could help reduce the extent of BMI increases in individuals with ID.
Ilkim and Akyol (2018) [55] **/** Türkiye	Male and female children and adolescents with DS (n = 20) EXP (n = 10) aged (12.5 ± 0.8) CON (n = 10) aged (13.6 ± 1.2)	EXP implemented program of table tennis. CON implemented program of throwing ball on a wall, hitting ball with foot, hitting the ball with the cuff, jogging, running.	To assess the reaction times of individuals with DS, participated in table tennis exercise program.	12 weeks	Reaction time test	EXP ↑↑ RT, NC CON RT. Children with DS who were included in the table tennis activity (3 times per week, 90 min) achieved better reaction time results compared to the children of CON.
Naczk et al. (2021) [56] / Poland	Male and female adolescents with DS (n = 22) male (n = 14) female (n = 8) EXP (n = 11) aged (14.9.5 ± 2.3) CON (n = 11) aged (14.4 ± 1.9)	EXP participated in 33 weeks of water-based exercise and a swimming program. CON maintained their normal daily activity.	To estimate the influence of a 33-week swimming program on aerobic capacity, muscle strength, balance, flexibility, and body composition of adolescents with DS.	33 weeks	Body composition assessment; physical fitness Eurofit test battery; aerobic capacity test; water orientation test Alyn 2	eXp ↓ BM, BF, BMI, CON ↑ BM, BF, BMI, EXP ↑ VO2max, AS, CON ↓ VO2max, EXP ↑ SAS, TS, EFS, NC EXP SOL, B, MSIT, NC CON AS, SAS, TS, EFS, SOL, FBT, MSIT. Swimming program is strongly recommended for people with DS.

Note: EXP, Experimental group; CON, Control group; ID, Intellectual Disability; IDID, Intellectual Disability; DS, Down syndrome; PE, Physical Education; SO, Special Olympics; UNS, Unified Sports Soccer; SSGs, Small-sided games; FAS, Friendship Activity Scale; BH, Ball handling; PS, Passing scores; R, Reception; SS, Shooting scores; LST, Leg strength test; WST, Wall squat test; DHGS, Dominant hand grip strength; NDHGS, Non-dominant hand grip strength; VJ, Vertical jump; VJAS, Vertical jump with arms swing; SBJ, Standing broad jump; SLJ, Standing long jump; SONL, Standing on one leg; MPUS, Push-ups; AS, Aquatic skills; SAS, Static arm strength; TS, Trunk strength; EFS, Endurance/functional strength; FBT, Flamingo balance test; RT, Reaction time; MSIT, Sit-and-reach; MHTG, Hand tapping; SOL, Speed of limb; PCR, 16m Pacer; SHD, One-minute single-handed dribble; OMS, One-minute shoot, MTDA, Throwing darts; AC, Athletic competence; SA, Social acceptance; PA, Physical appearance; GSW, General self-worth; FAH, Flexed-arm hang; MSAR, Sit and reach; PQ, Psychological questionnaires; MCUP, Modified curl-up RB, Receiving ball; BMI, Body Mass Index; BM, Body mass; BF, Percent body fat; H, Height; W, Weight; WH, Weight-height ratio, WC, Waist circumference; HC, Hip circumference; DP, Diastolic pressure; AP, Average arterial pressure; HR, Heart rate; RHR, Resting heart rate; 6MWT, Six-minute walk test; HC, Half court; FC, Full court;; IM, Initial measurement; FM, Final measurement; ES, Effect size; NC, No statistically significant changes *p* > 0.05; ±, Mean and standard deviation; ↑, Statistically significant increase *p* < 0.05; ↑↑, Statistically significant increase *p* < 0.01; ↓, Statistically significant decrease *p* < 0.05.

## 4. Discussion

The primary purpose of this review was to determine the effects of sports game programs on motor skills in children, adolescents, and youth with intellectual disabilities and DS. A review of research to date has found that sports game programs, particularly basketball, represent a safe and reliable way to exercise, and an effective and practical rehabilitation program for individuals with ID. Additionally, the college football program has a positive impact on the speed of concentric contraction of the leg-extensor muscles, and therefore, on the reflective impulse.

Studies consisting of basketball players clearly show the positive effects of basketball programs on the motor skills of children, adolescents, and youth with ID. The basketball test battery can be used to improve and monitor basketball training [32,47,59,64]. Basketball training has contributed to positive changes in the final measurement regarding improved ball handling (BH), reception (R), as well as passing scores (PS) and shooting scores (SS). Mohammadi and associates [32] clearly showed that the study participants who were given easy goals improved their free-throw shooting performance in basketball, compared to a group who were given difficult goals and did not improve their performance.

The results indicate that basketball players with lower levels of ID achieve better results, as the best results were obtained in basketball players with level II and III of ID [29,47], and significant differences were obtained between level categories (I, II, III) of ID in all fields [59]. Additionally, the explosive power of the legs contributes the most to the increase in ID levels II and III [63].

Studies consisting of students, adolescents, and adults also show the benefit of basketball programs on motor abilities [49,51] and muscular strength of the upper and lower extremities in people with ID [64]. However, Stanišić and associates [49] indicate that an eight-week specially designed basketball program contributes to an increase in specifically motor skills of adolescents but not physical fitness, and the key reason is probably the short time spent in training.

The study of Tsimaras and associates [64], whose training program lasted four years, indicated differences in all tests between males with and without ID. The experimental group that participated in the four-year basketball training had higher absolute and relative values of the extensors and flexors of both knees, while the experimental group that exercised recreationally had greater antagonistic activity of the extensors and flexors of both knees. It should be emphasized that both experimental groups have higher antagonistic activity of the extensors and flexors of both knees than the control group. Hemayattalab and Movahedi [65] examined the effect of five different variations of physical exercise on learning in adolescents with ID regarding throwing free throws. They indicated that it is cognitively related to physical exercise and significantly contributes to the success of free throws in people with ID. A specially adapted basketball training program also has a positive effect on the physical fitness of basketball players, especially in the term of heart rate (HR) and a six-minute walking test (6MWT), but without changes in anthropometric dimensions, thus providing limited information on the effects; these results support the design of a full-scale experiment on this topic [50]. Therefore, basketball is a safe and reliable way to exercise for people with ID because it implies sustained and continuous physical activity and is therefore recommended as an effective and practical rehabilitation program for individuals with ID. In addition, the futsal program implemented with individuals with ID contributed to improving the results of the vertical jump test, which is an indicator of the explosive power of the legs, but without statistically significant differences. However, that experimental program had an even more significant impact on the coordination, and the differences between the test results—a vertical jump with and without an arm swing, which can be seen indirectly as a coordination factor—were statistically significant. The results of this research recommended futsal as a very interesting and helpful activity for individuals with ID [62].

Studies conducted by the SO UNS program are helpful for young people with and without ID, as it reduces behavior problems, social competence (SC), and the Friendship Activity Scale (FAS) in conjunction with physical education classes; in addition, it improves the relationship of youth without ID to participants with ID [63]. Baran and associates [60] point to the importance of the UNS program eight weeks after full-time physical education, which shows significantly higher physical fitness and soccer skills in males with and without ID compared to a control group that did not participate in any sport after regular physical education classes. It should be emphasized that there were no differences in behavior and interaction between groups and that the experimental group that had basketball SO training and school competitions performed better than the experimental group that had SO running training and school competitions [57]. Thus, SO training and the UNS program successfully increase the fitness abilities and performing abilities of football skills and reduce behavioral problems, SC, and FAS. The positive effect of SO Fit 5 training on resting heart rate (RHR), blood pressure, and BMI in adolescents and adults with ID was confirmed in the study [48].

In addition to those with ID, those included were also individuals with DS, which will be explained in more detail in the following section. Cai and Baek [54] obtained results that clearly show the positive effects of a 24-week basketball program on improving body composition, flexibility, balance, aerobic capacity, and basketball functional abilities in individuals with DS. The positive effects of the basketball program were confirmed by Aydogan and Demirok [61], who investigated the effects of basic basketball movements on the effectiveness and persistence of video modeling lessons. The results of the study showed that teaching by video models is an effective method in providing basic basketball skills to adolescents with DS.

It should be emphasized that no additional studies have been found that examine the effects of (collective) sports games programs on motor skills in children, adolescents, and youth with DS. On the other hand, there are studies that confirm the positive effects of swimming and table tennis programs in children and adolescents with DS. Ilkim and Akyol [55] showed that the reaction times of children with DS who participated in table tennis activities were better in comparison to the control group. Naczk and associates [56] indicated that a thirty-three-week swimming program has positive effects on muscle strength, aquatic skills, and health status in adolescents with DS.

Ince [52] emphasized that an eight-week BH program contributed to a statistically significant increase in upper and lower extremity muscle strength in adolescents and adults with DS as well as an increased leg strength test (LST), wall squat test (WST), and dominant handgrip strength (DHGS); however, there were no changes in the vertical jump (VJ), standing broad jump (SBJ) and non-dominant handgrip strength tests (NDHGS). It is interesting to note that there were no differences between groups, although the control group did not participate in any of the activities. In addition to the positive effect on motor skills or muscle strength, the positive effect of twelve-week training on anthropometric characteristics, body composition, blood fat, and blood pressure in individuals with and without DS did not contribute to the reduction in low-density lipoproteincholesterol [66]. Other studies confirm the positive effects of exercise programs on muscle strength in children with DS [67,68]. Therefore, muscle strength is a significant fitness parameter that needs to be developed, which means that further studies are needed to find possible factors that would contribute to the increase in muscle strength in children, adolescents, and youth with DS.

## 5. Conclusions

This is the first review study to investigate whether there are positive effects of sports game programs on motor skills in individuals with ID and DS. Basketball is recommended as an effective and practical rehabilitation program for people with ID and DS, including fitness parameters, motor skills, and interaction aspects. Basketball players with lower levels of ID achieve better results, especially those with ID II and III degrees, especially regarding better fitness abilities. However, some disagreements can certainly be attributed to the individual characteristics of the sample of participants. SO training and the UNS program successfully increase the fitness abilities and performance of soccer skills, reduce behavioral problems, SC, and FAS, which once again, confirmed the importance of basketball and futsal sports games, either in young people with or without ID. In individuals with DS, training programs contribute to a statistically significant increase in upper and lower extremity muscle strength; additionally, positive effects on anthropometric characteristics, body composition, blood fat, and blood pressure should be emphasized. Further studies are needed to investigate the possible factors that would contribute to the increase in muscle strength in those people.

## Figures and Tables

**Figure 1 children-10-00912-f001:**
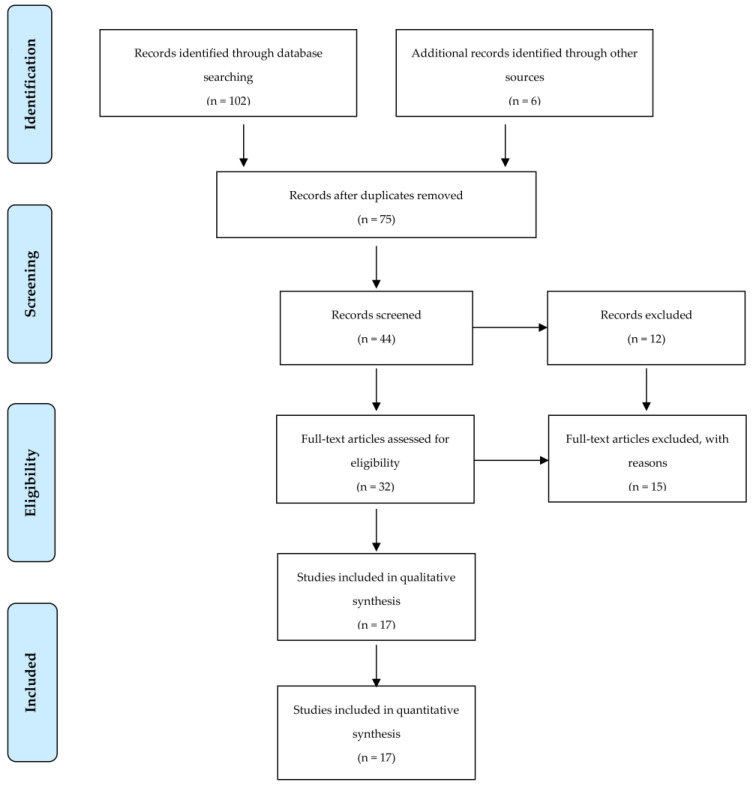
Flow chart diagram of the study selection.

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
