# Peer review of "Sports Games and Motor Skills in Children, Adolescents and Youth with Intellectual Disabilities"

_children, 2023, doi:10.3390/children10060912_

Round 1

Reviewer 1 Report

Dear authors,

Thank you for your contribution. The objective of your study is clearly defined, but the results and discussion need to be improved.

The aim of your study was to present the effects of sport games on motor skills in children, adolescents, and youth with intellectual disabilities and Down syndrome, but the review presented only the results of two articles related to Down syndrome: references 59 (Ince) and 61 (Cai). I suggest analyzing more articles related to the effects of physical activity in children, adolescents, and youth with Down syndrome.

Major revision:

-        The results must present in detail the groups analyzed in each study (subjects with or without MR/ID, type of intervention for each subgroup, assessment tools used in each study, what does EXP, EXP1, EXP2, CON, CON1, CON2 represent) 

-        Were the effects of sport games on subjects with DS and ID statistically significant correlated with the ID level?

-        Structuring the results according to the key findings for each monitored parameter would highlight the benefits of sports games on subjects more than actual structure, respectively, according to the study author; if the monitored parameters are (if the monitored parameters are totally different, they could be grouped, for example: muscle strength, precision, execution speed, etc.). 

Minor revision:

-        Table 1: Kasum is reference 54 and Radenkovic 57

-        Table 1: Ince study (reference 59) consists of 21 participants, 10 EXP and 11 CON

-        Please define in the Introduction section the ID levels

-        Please clarify if this research received funding or not (lines 349 and 352)

-        Please correct the Author Contributions (lines 342-351)

Author Response

Dear,

We would like to express our gratitude to the reviewer for their invaluable contribution in significantly improving our manuscript.

Please find our responses to the reviewer's comments attached herewith.

Thank you and regards,

Authors

Reviewer 2 Report

This systematic review aimed to identify relevant data on motor skills and to clarify whether there are positive effects of sports programmes on motor games in children, adolescents and young adults with intellectual disabilities and Down syndrome. It is a great contribution to this group due to the lack of studies available to advance inclusion.

I think that the article with a bit more attention to the wording and language. Congratulations

Se requiere una edición menor del idioma inglés

Author Response

(The authors gave the same response as above.)

Reviewer 3 Report

Pertinent theme and lacking systematization of studies like this one. The authors substantiate the rational well and their methodological procedures are adjusted and clear. With regard to the results, it is suggested that in table 1 an indication of the country of the study appear with the authors. The conclusions are clear and result from the evidenced results.

Author Response

(The authors gave the same response as above.)

Round 2

Reviewer 1 Report

The manuscript has undergone significant improvements, with enhanced clarity and consistency in presenting the results, conclusions, evidence, and arguments. Additionally, the introduction has been strengthened by incorporating essential explanations.

However, there is one minor issue that needs to be addressed: clarification regarding the funding status of this research. In line 381, it states, "Funding: This research received no external funding." However, on line 377, it mentions "funding acquisition, T.D.; A.-M.V." This inconsistency should be rectified to provide accurate information regarding the funding sources for this study.